# One-class Gaussian process regressor for quality assessment of transperineal ultrasound images

**Saskia M. Camps***
Faculty of Electrical Engineering
Eindhoven University of Technology
Eindhoven, the Netherlands
s.m.camps@tue.nl

**Tim Houben***
Faculty of Electrical Engineering
Eindhoven University of Technology
Eindhoven, the Netherlands
t.houben@student.tue.nl

**Davide Fontanarosa**
School of Clinical Sciences
Queensland University of Technology
Brisbane, Queensland, Australia
d3.fontanarosa@qut.edu.au

**Christopher Edwards**
School of Clinical Sciences
Queensland University of Technology
Brisbane, Queensland, Australia
c8.edwards@qut.edu.au

**Maria Antico**
SCPME
Queensland University of Technology
Brisbane, Queensland, Australia
maria.antico@hdr.qut.edu.au

**Matteo Dunnhofer**
DMCSP
University of Udine
Udine,Italy
mattdunnhofer@gmail.com

**Esther G.H.J. Martens**
MAASTRO
GROW
Maastricht, the Netherlands
esther.martens@maastro.nl

**Jose A. Baeza**
MAASTRO
GROW
Maastricht, the Netherlands
jose.baeza@maastro.nl

**Ben G.L. Vanneste**
MAASTRO
GROW
Maastricht, the Netherlands
ben.vanneste@maastro.nl

**Evert J. van Limbergen**
MAASTRO
GROW
Maastricht, the Netherlands
evert.vanlimbergen@maastro.nl

**Peter H.N. de With**
Faculty of Electrical Engineering
Eindhoven University of Technology
Eindhoven, the Netherlands
p.h.n.de.with@tue.nl

**Frank Verhaegen**
MAASTRO
GROW
Maastricht, the Netherlands
frank.verhaegen@maastro.nl

**Gustavo Carneiro**
Australian Centre of Visual Technologies
the University of Adelaide
Adelaide, Australia
gustavo.carneiro@adelaide.edu.au

31st Conference on Neural Information Processing Systems (NIPS 2017), Long Beach, CA, USA.

## Abstract

The use of ultrasound guidance in prostate cancer radiotherapy workflows is not widespread. This can be partially attributed to the need for image interpretation by a trained operator during ultrasound image acquisition. In this work, a one-class regressor, based on DenseNet and Gaussian processes, was implemented to assess automatically the quality of transperineal ultrasound images of the male pelvic region. The implemented deep learning approach achieved a scoring accuracy of 94%, a specificity of 95% and a sensitivity of 93% with respect to the majority vote of three experts, which was comparable with the results of these experts. This is the first step towards a fully automatic workflow, which could potentially remove the need for image interpretation and thereby make the use of ultrasound imaging, which allows real-time volumetric organ tracking in the RT environment, more appealing for hospitals.

# 1 Introduction

One of the treatment modalities for prostate cancer is radiotherapy (RT). This modality aims to irradiate tumor tissue, while sparing the surrounding organs at risk (e.g. bladder and rectum) as much as possible. The radiation dose is typically delivered to the patient in multiple treatment fractions in accordance with a treatment plan designed based on a computed tomography (CT) scan.

It has been shown that the shape and position of the prostate might differ between treatment fractions (inter-fraction), due to change in bladder and/or rectal filling [1]. Also during a treatment fraction (intra-fraction) the tissue distributions might change [2]. If the original treatment plan was delivered on the changed tissue configuration, this could potentially result in a suboptimal dose deposition in the tumor and the organs at risk could receive extra undesired dose [3].

For this reason, several solutions have been proposed to identify the (relative) position and shape differences of the anatomical structures during the treatment course with respect to the treatment plan. This information can be used to potentially improve dose delivery precision. Most of the proposed solutions require frequent imaging during the course of the RT treatment (image guided RT, IGRT) with or without implanted fiducial markers [4] using X-ray, magnetic resonance imaging (MRI) [5, 6] or ultrasound (US) imaging [7, 8, 9].

In this work, we focused on the use of US imaging for intra-fraction guidance during RT (USgRT). US imaging allows real-time volumetric organ tracking in the RT environment and in addition, it is relatively cheap and it is harmless for the patient. Despite these advantages, its use is not yet widespread. This can be partly attributed to the need for a trained operator during manual image acquisition to verify if the correct anatomical structures are visualized with sufficient quality.

To allow for intra-fraction monitoring of anatomical structures, the operator needs to position the US probe prior to the fraction. As the operator cannot stay in the treatment room during radiation delivery, the probe would need to be fixed using either a mechanical or a robotic arm. During the treatment fraction, small motion of the patient or changes in anatomical structures can compromise image quality due to, for example, a loss of acoustic coupling and/or a sudden appearance of shadowing artifacts. The operator would therefore need to be present in the control room to promptly identify this quality loss and, if necessary, take appropriate action.

The aim of this study was to develop a prototype deep learning algorithm to automatically score 2D US images of the male pelvic region based on their quality or, in other words, on their usability during the USgRT workflow. More specifically, we achieve this aim with the development of a novel one-class regressor, based on DenseNet [10] and Gaussian processes (GPs) [11]. This is the first step towards a fully automated workflow that would remove the need for a trained operator and therefore potentially make the use of US imaging more appealing for hospitals.

Machine learning has been used before for quality assessment of US images, primarily in the obstetrics field (e.g. [12, 13]). In these studies, the quality assessment was based on initial segmentations or on the presence of specific anatomical structures in the image. Instead, we, aim at performing the quality assessment using solely automatically learned deep features from the image without relying on any initial segmentations or specific anatomical structure detection.

## 2 Materials and methods

### 2.1 Image data acquisition

In this work, datasets from three different studies on 36 male subjects were combined (Table 1). The subjects were either healthy volunteers or patients with localized prostate cancer. For each subject, several 3D and 4D transperineal US (TPUS) volumes of the pelvic region were acquired using an X6-1 xMatrix array probe (Philips Healthcare, Bothell, WA, United States) and an EpiQ7 US system (Philips Medical Systems, Andover, MA, USA). The datasets show a variability in image characteristics due to:

1. different volume dimensions and voxel sizes, due to a requirement to achieve an acceptable frame rate in the 4D sequences;

2. varying body composition, age and medical history of the subjects;

3. possible anatomical structure displacements, which were artificially introduced by instructing the subjects to consciously contract muscles in the pelvis area or to cough;

4. the exact settings on the US system such as imaging depth and focus, which could vary between the different studies and between the different subjects (this also affected the voxel sizes and dimensions of the volumes);

5. the involvement of four radiation oncologists in the acquisition of the volumes, each of them with their own approach to US image acquisition;

Table 1: Summary of the available datasets in this study in total comprising 11,148 TPUS volumes from 36 male subjects

| Study | Subject type | # subjects | Age [mean years] | Total # volumes |
|-------|-------------|-----------|------------------|-----------------|
| **Study 1** | Volunteers | 6 | 35 (range: 26-52) | 840 |
| **Study 2** | Patients | 21 | 74 (range: 58-85) | 1,269 |
| **Study 3** | Volunteers | 9 | 51 (range: 31-73) | 9,039 |
| **Total** | - | 36 | - | 11,148 |

### 2.2 Initial image data pre-processing

Three initial pre-processing steps were necessary to prepare the datasets for deep learning algorithm processing. These steps were all performed using MATLAB (Version 9.3.0 (R2017b), The Mathworks Inc. Natick, MA, United States). First, the volumes were resampled to identical voxel sizes, which allowed easy volume comparisons and batch processing of the data in the next steps. Second, the TPUS volumes were sliced to 2D images along the axial direction, as this was the direction with the highest resolution. Visual inspection of these images then revealed that the anatomical structures of interest were most often located in the center of each volume. For this reason, only the central 16 axial 2D images from each volume were selected for further processing, which also reduced the total computational cost. Then all 2D images were symmetrically padded with black pixels to ensure that all images had the same dimensions as the largest 2D image (216x180 pixels) in the entire dataset. Finally, a fixed region of interest was defined by cropping the images, while preserving the crucial information of all anatomical structures. This resulted in 178,368 2D TPUS images composed of 116x100 pixels originating from 11,148 TPUS volumes.

### 2.3 2D US image classification

The crucial anatomical structures for prostate RT treatments are: prostate, seminal vesicles, bladder and rectum. Prostate is the target of the treatment and should therefore be completely visible on an acquired US volume. In the ideal case, also the edges of the bladder and rectum adjacent to the prostate should be visible to be able to potentially spare these organs at risk from excessive radiation exposure. As it was not possible to identify the seminal vesicles with sufficient certainty on the acquired US volumes, these were not evaluated in this study.

Based on the above-mentioned criteria, three image categories were defined which are detailed in Table 2. An example of each category is displayed in Fig. 1. Category 1 involves images that have insufficient quality to be used clinically for USgRT. The quality of Category 2 and 3 images was considered sufficient as the target is visualized and potentially can be tracked.

Table 2: Definition of three image criteria used to classify 2D TPUS images based on their quality.

| Category | Criteria |
|---|---|
| **Category 1** | Prostate could not be identified |
| **Category 2** | Prostate or prostate and also a part of bladder or rectum could be identified |
| **Category 3** | Prostate could be identified, as well as a part of bladder and rectum |

In order to provide the deep learning algorithm with labeled training, validation and test samples, a subset of the available 2D TPUS images was manually and independently scored by four members of our research team, as the experts involved in this study had very limited time available. The central 16 2D images (see Section 2.2) of each volume were presented to each team member. They could then scroll through the images of each volume and assign a score between 1 and 3, corresponding to Categories 1 to 3, respectively, to each image. Some of the 2D images were horizontally flipped, due to the fact that the probe was sometimes held upside down. This resulted in a mirrored anatomical structure configuration. During the scoring process, the orientation of these images was manually corrected, to ensure that the bladder was located on the left side and the rectum on the right. The team members were instructed to only assign a score to an image if they were highly confident, so it was also possible to leave some images unscored. Following this procedure, 1000 randomly selected volumes were scored by each team member.

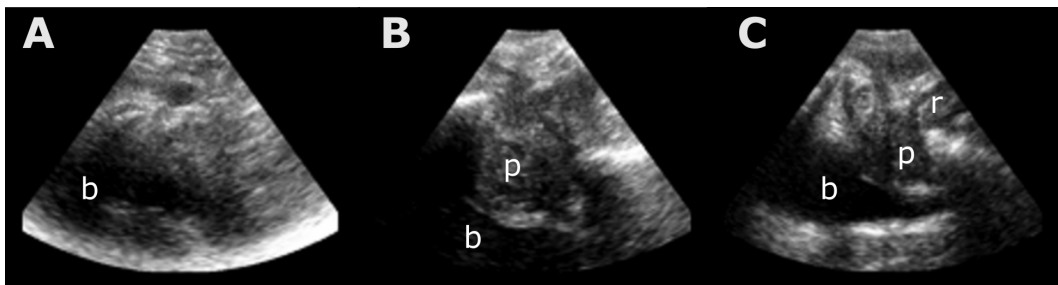

Figure 1: Example 2D US image of each quality category. (A) Category 1 with only bladder (b) visible; (B) Category 2 with bladder (b) and prostate (p); (C) Category 3 with bladder (b), prostate (p) and rectum (r).

The images that received the same score from at least three out of four team members were included in a database (*Database_NonBinary*) with the majority vote of the scores given by the team members assumed to be their ground truth annotations. Subsequently, the scores of each team member were binarized, with binary score 1 = 0 and binary score 2 or 3 = 1. Then, the same procedure of selecting the majority vote of the scores of all team members was followed, resulting in a binary database (*Database_Binary*).

## 2.4 Subject data split and database generation

The research team evaluated overall 16,000 2D TPUS images distributed over 1,000 volumes. In total 13,463 of these images (from 34 out of 36 subjects) received the same binary score from three out of four team members and were therefore included in *Database_Binary*. Subsequently, the data was split into training (60%), validation (20%) and test (20%) sets. As mentioned in the previous section, the quality of Category 2 and 3 images was considered sufficient for use in clinical practice. Therefore, *Database_Binary* was used to train and test the algorithm; however, the subject split into training, validation and test sets was performed based on *Database_NonBinary*. This was done to ensure a balance between good (Category 2) and very good (Category 3) images in the positive binary group.

Not for all subjects images of all categories were available and, in addition, the number of classified images per subject also varied. For this reason, the split was not just done randomly, but performed using an optimization approach based on simulated annealing [14].

First, the data was split in a test and train set by randomly assigning the subjects to one of the groups, while not exceeding the defined sizes of each group. Subsequently, in each iteration, a random subject from the test set was swapped with a random subject from the training set. The aim was to obtain similar ratios between the number of images of a certain category (1-3) in each group (test or train) with respect to the total size of that group. In total 1000 iterations were executed, in which more weight was put on the ratios of Category 2 and 3 images. The same process was repeated to extract the validation set from the training set. In the end, this resulted in a distribution of bad quality images (binary score 0) and good quality images (binary score 1) over the train, validation and test set as visualized in Fig. 2A. In Fig. 2B the distributions of the binary score 0 and 1 images per subject and per group are detailed.

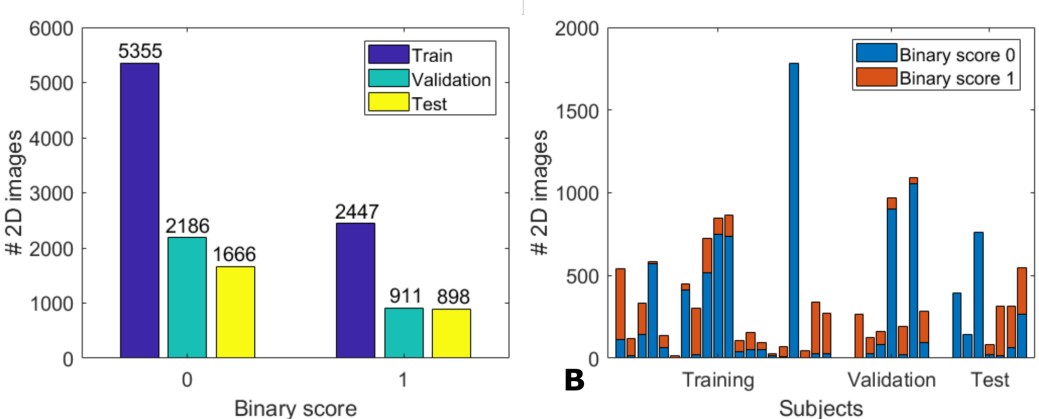

Figure 2: Distribution of the image data per binary score (A) and per subject (B) in the training, validation and test set.

## 2.5 Classification cross-validation

Cross-validation was performed by an accredited medical sonographer and two of the radiation oncologists who were involved in the acquisition of the images. These experts were each presented with the same 100 2D TPUS images, which were randomly selected from the test set of *Database_Binary* and asked to score these images between 1 and 3. The inter-expert agreement, the test data agreement and the performance of the algorithm were then compared to the majority vote of the experts using Fleiss' kappa [15], accuracy, sensitivity and specificity metrics.

## 2.6 Machine learning algorithm selection

Several different aspects of the image acquisition procedure as well as the patient body composition affect US image quality. Quality decrease can be caused by many factors including, insufficient acoustic coupling between the US probe and the skin, bones causing shadowing artifacts on critical anatomical structures and insufficient penetration due to (fat) tissue distributions. This makes describing features for classification challenging.

For this reason, we approached this problem as a one-class classification (OCC) problem. This approach involves the definition of a single class that should contain all images with "good" (according to clinical requirements) quality, while considering the images with "bad" (according to clinical requirements) quality as outliers. One-class support vector machines (OCSVM) can construct a hyper-sphere with a minimum radius, which contains all positive data points in the multidimensional feature space [16]. However, even though this technique is widely used, it does not perform well on noisy data and requires manual definition of regularization and kernel parameters [17].

In this work, the use of Gaussian processes (GPs) instead of conventional SVM was explored for OCC of US image quality. In line with Kemmler *et al.* [11], GPs were used for regression acting as a one-class classifier. In contrast to SVMs, GPs deliver probabilistic predictions and is able to automatically learn regularization and kernel parameters as well as feature importance. However, GPs lack characterization power for complex data [18]. For this reason, a combination of two techniques was considered: a convolutional neural network (CNN) was used as an autonomous feature descriptor, then feeding its output into the GP for OCC.

## 2.7 Architecture and implementation

In this work, DenseNet [10] was used for feature description. This CNN provides a robust architecture which reduces the chance to over-fit and the chance for vanishing gradients, while giving state-of-the-art results on fundamental datasets, like ImageNet [10]. A wide-sense variant with 2 dense blocks, a depth $L$ of 26 layers and a growth rate $k$ of 12 (see Table 3) were used and no bottleneck layers were included. Prior to the first dense block, a convolutional operation with a 7x7 filter was performed, followed by a max pooling operation. Finally, the last fully connected layer was removed and replaced by a GP regressor (see Fig. 3).

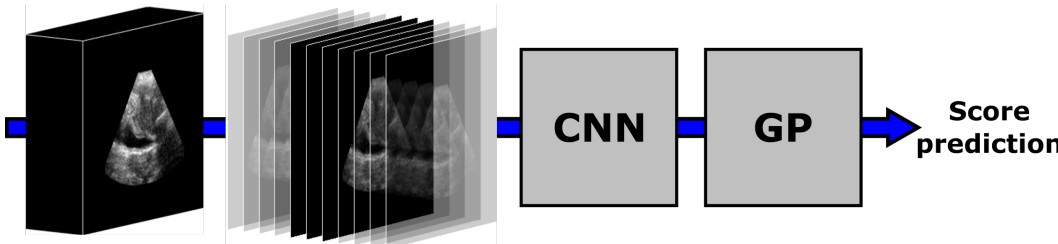

Figure 3: Machine learning approach. Selection of 2D images from the center of a TPUS volume, followed by processing using a CNN and a GP, resulting in a quality score prediction.

This regressor was implemented using GPflow [19]. A major advantage of GPflow is that it supports sparse GPs [20], which reduces computation time and memory usage (one of the main drawbacks of GPs). The regressor used a radial basis kernel function (RBF) with an initial variance of 0.1 to fit the data (see Table 3). The number of points used during the GP calculations was 100 which was about 65% of the outputs from the CNN. As the GPflow library is built on TensorFlow [21], the DenseNet was also implemented in TensorFlow to make end-to-end training possible.

Table 3: Algorithm parameter per implementation step with the asterisks indicating the optimized hyper-parameters.

|  | Parameter | Value |
|---|---|---|
| **DenseNet** | Depth $L$* | 26 |
|  | Growth rate $k$* | 12 |
|  | Outputs* | 156 |
| **GPflow** | Model | Sparse GP Regression (SGPR) |
|  | Kernel | Radial Basis Function (RBF) |
|  | Kernel variance | 0.1 |
|  | Inducing points* | 100 |
| **Training** | Batch size* | 150 |
|  | Epochs* | 100 |
|  | Optimizer | Adam |
|  | Learning rate* | 0% - 33%: 1e-5 |
|  |  | 33% - 67%: 1e-6 |
|  |  | 67% - 100%: 1e-7 |
|  | Drop-out rate* | 0.1 |

Prior to providing the deep learning algorithm with the image datasets, two final processing steps needed to be performed. First, all pixel values were normalized by setting the total mean to zero and the standard deviation to unity, to ensure that the training backpropagation algorithm of the CNN would work efficiently. Second, the training data was randomly permuted and then split in mini-batches to ensure subject balance in the mini-batches.

Finally, during training (150 images per batch), the algorithm was only provided with images with good quality (binary score 1). The optimization was done using the Adam optimizer [22] and an annealing schedule was used for the learning rate. This rate was decreased by one order of magnitude at 33% and 67% of the training time (see Table 3). After the deep learning hyper-parameters were optimized (indicated with an asterisk in Table 3), the training and validation sets were combined into the final training set.

## 3 Results

### 3.1 Network performance

The Fleiss' kappa among the three experts was equal to 0.92, while the kappa among the three experts and the test subset was equal to 0.91. The accuracy, sensitivity and specificity results with respect to the majority vote of the experts are detailed in Table 4. The accuracy on the test subset with respect to the majority vote was 95%, while the accuracy from the experts ranged within 97% - 99%. The test subset had the lowest sensitivity (88% compared to 95%-100%), but a specificity of 100%.

Table 4: Accuracy, sensitivity and specificity (1-false positive rate) results for the algorithm, test subset and three experts calculated with respect to the majority vote of the three experts.

|  | Algorithm | Test set | Expert 1 | Expert 2 | Expert 3 |
|---|---|---|---|---|---|
| **Accuracy** | 94% | 95% | 97% | 98% | 99% |
| **Sensitivity** | 93% | 88% | 100% | 95% | 97% |
| **Specificity** | 95% | 100% | 95% | 100% | 98% |

In Fig. 4 the receiver operating characteristics (ROC) curve of the algorithm is plotted, again with respect to the majority vote of the experts. The green diamond indicates the highest accuracy of the algorithm (94%), which corresponded to a sensitivity of 93% and a specificity of 95% (see Table 3). The red, blue and magenta diamonds indicate the performance of the experts, while the cyan diamond corresponds to the test subset. The Fleiss kappa for the experts and the algorithm was equal to 0.9.

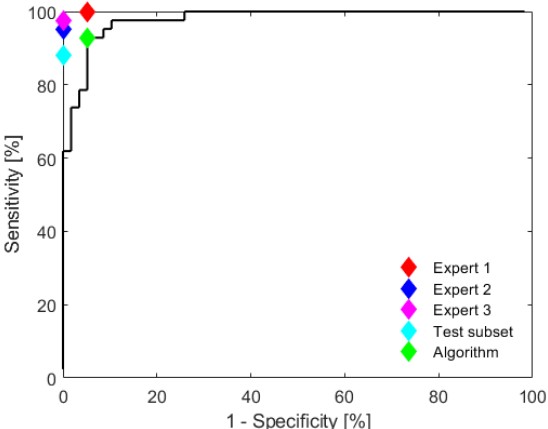

Figure 4: ROC-curve of the algorithm with respect to the majority vote of the experts, where the red, blue and cyan diamonds are indicating the performance of the experts, the cyan diamond gives the performance of the test subset and the green diamond gives the performance of the algorithm.

# 4 Discussion

In this work, a deep learning approach was proposed that can be used to automatically assess the quality of TPUS images of the male pelvic region. The algorithm was trained based on a subset of a larger database and the labels used for training were generated by a small team. The team members were asked to only assign a score when they were highly confident and only images to which at least three out of four members assigned the same score were included in the database. This was done to eliminate part of the inter-user variability. In the ideal case, the labels would be generated by experts, but this was not feasible due to time constraints. However, the kappa values (0.92 vs. 0.91) showed a good agreement between the team-members and the experts (see Table 3). The accuracy of the test subset was 95%, which is lower than the accuracy of the experts, but still comparable.

The aim was to achieve an accuracy of 95% with the algorithm as well. In Fig. 4 it can be observed that the algorithm is able to achieve a sensitivity and specificity that are comparable with the experts, which resulted in an overall accuracy of 94%. Calculating the Fleiss kappa of the experts and the algorithm resulted in 0.9, which seems to imply that there is almost perfect agreement (according to the interpretation of Fleiss' kappa from Landis and Koch [23]). The current performance evaluation was performed with a subset of the test set. In future research, this subset will be expanded and the algorithm parameters will be optimized further in order to achieve the 95% accuracy goal.

The scores assigned to the images were binarized, as the quality of Category 1 was considered insufficient for use in clinical practice, while the quality of Category 2 and 3 was considered sufficient. In Category 2 images, none or just one of the organs at risk is visualized. As these organs should be spared from radiation as much as possible, in the future not only the position of the prostate could be monitored, but also the position of these organs. This would introduce the need to also make a distinction between Category 2 and 3 images. In addition, a single bad-quality 2D image does not necessarily imply that the whole volume is not able to provide useful clinical information. Therefore, the next steps should be to move towards the interpretation of a whole volume, for example, using recurrent neural networks which can take into account inter-slice context (e.g. [24]).

The potential of the database that was available in this work has not been fully exploited, as only 16,000 2D images of the 178,368 images were examined by the team, resulting in 13,463 images with labels. Potentially, the performance of the algorithm can be improved by using more images for training. Also, the orientation of the images that had a mirrored anatomical structure configuration were manually corrected during the scoring process. However, during the actual image acquisition the probe might be held upside down as well, so the algorithm should be robust for any image orientation changes. This robustness will also be examined in future research.

The focus in this work was on the use of US imaging during the RT workflow of prostate cancer patients. However, a similar approach could be adapted for use in other medical procedures in which US imaging may be beneficial for anatomical localization, but it is not feasible and/or desirable to have a trained operator present at the time. These procedures could be, for example, USgRT workflows of other cancer sites (e.g. liver, bladder of cervical cancer) or US guided surgeries.

# 5 Conclusion

The purpose of this work was to propose a deep learning approach that could be used to automatically assess quality of TPUS images of the male pelvic region. This could potentially remove the need for quality interpretation by a trained operator. The performance of the implemented one-class Gaussian process regressor was compared with three experts and the results showed that the algorithm is comparable with these experts in a binary scoring scenario. Future work will involve exploring the non-binary scoring scenario, including adding additional data into the database and assessing the overall quality of the TPUS volume instead of judging individual 2D images.

**Acknowledgments**

The computational resources and services used in this work were provided by the HPC and Research Support Group, Queensland University of Technology, Brisbane, Australia. This research forms part of a project supported by an Australia-India strategic research fund (AISRF53820). G.C. acknowledges the support received by the Australian Research Council's Discovery Projects funding scheme (project DP180103232).

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
