# OpenReview forum: "One-class Gaussian process regressor for quality assessment of transperineal ultrasound images "
_MIDL.amsterdam/2018/Conference — MIDL 2018 Poster_

### Review · AnonReviewer2 · 2018-05-01
**This work proposes to use a CNN with a GP head to predict image quality for UGRT, using one-class classification. The clinical motivation is strong. In particular, the hypothesis that learning from expert’s scores to quantify imaging quality for ultrasound guidance is both interesting and promising. The label data are impressive. However, the motivation of the methodology especially the one-class formulation is not clearly explained, while the validation experiment may be questionable.**

**Rating:** 2
**Confidence:** 3

**Review:**

The variation of negative class seemly motivated the adoption of one-class classification in this work, as in “Quality decrease can be caused by many factors including, insufficient acoustic coupling between the US probe and the skin, bones causing shadowing artifacts on critical anatomical structures and insufficient penetration due to (fat) tissue distributions. This makes describing features for classification challenging.” - This may not be appropriate. One-class is usually becoming interesting if the training data size of the negative class is limited, which is not the case in this study (i.e. Figure 2). Being driven by large variation of certain class might be intuitive, but conceptually, two-class classification only needs to find the decision boundary, while one-class needs to approximate the distribution of the class of interest, which is often considered as a harder problem that may require more data.
Secondly, from an algorithmic perspective, the benefit is not clearly demonstrated by training of the CNN with a “GP layer” in the end, compared with, for instance, a pure CNN predicting either one- or two class probabilities. Especially when training end-to-end as described in the paper, the CNN or a deeper variant should be able to provide equivalent expressibility, while the added value of the additional architecture constraints from GP is not obvious.
I would suggest that extra results to support the proposed method would be important, from a comparison between CNNs with- and without GP and those minimising two-class classification losses with negative class data in training.

I also have my reservation in the sampling scheme adopted in the validation experiment. If obtaining a comparable class ratio needs explicit optimisation, is it a good indication that either training or validation/test data does not have adequate sample size? Especially on the patient level, the reported results are likely to bias towards the 36 patients data set, with questionable generalisation ability. This effect could be investigated by using simple random sampling, which should represent different class distributions (both in training and test/validation), ones that are likely to have during generalisation, and see how the results differ from that reported in the paper.

A few minor points:
Reslice images may not be representative for 2D slices obtained during RT, especially when applied with a 2d probe. If applied with 3D probe, why is 2D quality important? A detailed application may explain this better for the readers.

The justification of including healthy subjects for an prostate cancer application is not clear.

“Flipped” images due to probe rotation do not have “mirrored” anatomy.

The reason to use the Adam optimiser with an annealing scheme is not clear.

**Special Issue:**

No

---

### Review · AnonReviewer1 · 2018-05-09
**The paper describes an approach to automatically detect the presence of certain structures in ultrasound images. The clinical motivation is poorly motivated and not convincing. The presented results appear to be good, however, some technical aspects are not well considered.**

**Rating:** 3
**Confidence:** 3

**Review:**

The paper describes an approach to automatically detect the presence of certain structures (prostate, bladder, rectum) in ultrasound images used for localization in radiation therapy. An impressive number of labeled data is used to train a dense neural network. While the approach yields good results on the presented data, some of the technical aspects remain vague, others are questionable.
There is previous work on the assessment of image quality on ultrasound images. Some of it is mentioned in the text, while other are missing (e.g. Schwaab, et al. J Med Imaging (Bellingham). 2016 Apr; 3(2) and others). This is a challenging task and the paper describes a valid approach, which yields good results (comparable to experts).
It is a bit farfetched to talk about “image quality” when “just” the presence of some organs or structures is analyzed. It would help to provide a more descriptive title, which better captures the main purpose.

Pros:
-	Nice utilization of an artificial neural network to judge the presence of certain structures in ultrasound images.
-	A huge amount of datasets.
-	There is a clear need for the solution of the described medical problem
Cons:
-	The clinical motivation for this particular approach is not very strong. The hypothesis that ultrasound is not used more often in RT because of the user dependent application, is not convincing. It does not take long for a technologist to learn how to position the ultrasound probe to make sure the prostate (and bladder and rectum) is in the imaging volume (supported by the fact that the authors write that it turned out that the relevant structures were always in the center of the scans.
-	The labeling process is somehow questionable. As the authors write, four members of the research team (presumably no medical experts, as this is nowhere stated) tagged the images for existence of prostate, bladder and rectum. Maybe, they were originally trained by a single person, who told them the characteristics of ultrasound images. I don’t think that this yields truly independent labels.
-	Little justification is provided why 2D images were assessed. In my opinion, it would be more straightforward to perform the analysis on 3D patches.
-	Even though the total number of datasets is impressive, the number of different patients/subjects is not. Due to the rather limited number a bias might arise.
-	I am not an expert in Gaussian proceses, but it didn’t become clear to me why this approach is the best. It would be nice to explain this is some more detail.


**Special Issue:**

No

---

### Review · AnonReviewer3 · 2018-05-10
**Method is indeed interesting; but missing comparison or even ablation study is what keeps me away from a stronger recommendation.**

**Rating:** 3
**Confidence:** 2

**Review:**

This paper presents a hybrid deep learning / Gaussian process-based method for detecting "reasonable" ultrasound images for prostate cancer radiotherapy. It takes a "one-class" rather than classification approach. Performance is demonstrated on a new dataset formed by merging several studies (36 subjects total and 11,148 volumes).

Strengths:
- Paper is very clear about the data collection strategy, assembly, preprocessing, etc.
- Interesting approach which combines the benefit of convnet-based feature extractors and GPs
- System is trained completely end-to-end
- Paper is also very clear about reporting all hyper-parameters for reproducibility

Weaknesses
- No comparison to other methods or even an ablation study
- The test set is very small
- The motivation for taking the one-class over classification approach is ok; but why not evaluate the technique as a classifier as well; this would just require a simple modification in code and re-training?

- The way that experts evaluated a subset of the test set since labeling was done by non-experts was interesting
- You use a very expensive procedure for forming the dataset split: 1,000 iterations of SA are required; is this really important?
- I don't know that "[SVM]... requires manual definition of regularization and kernel parameters" is a serious criticism; this could be said of the GP. The convnet has many more hyper-parameters as well.

Minor comments:
- Check Category 2 line in Table 2 "Prostate or prostate and..."


**Special Issue:**

No

---

### Decision · Program_Chairs · 2018-05-15
**Paper73 Acceptance Decision**

Poster